# Assessing Rodent Attitudes: The Psychometric Properties of the SARod in a Chilean Context

**DOI:** 10.3390/ani14223239

**Published:** 2024-11-12

**Authors:** Beatriz Pérez, Àlex Boso, Mauricio Herrera, Boris Álvarez, M. Cecilia Castilla

**Affiliations:** 1Department of Psychology, Universidad de La Frontera, Temuco 4811322, Chile; 2Department of Psychology, Universidad de Oviedo, 33003 Oviedo, Spain; 3Department of Environment, Centro de Investigaciones Energéticas, Medioambientales y Tecnológicas (CIEMAT), 08193 Cerdanyola del Vallès, Spain; alex.boso@ufrontera.cl; 4Department of Social Sciences, Universidad de La Frontera, Temuco 4811230, Chile; b.alvarez01@ufromail.cl; 5Faculty of Natural Resources and Veterinary Medicine, Universidad Santo Tomás, Puerto Montt 5504749, Chile; mherrera30@santotomas.cl; 6Centro Regional de Energía y Ambiente para el Desarrollo Sustentable, CONICET-UNCA, San Fernando del Valle de Catamarca 4700, Argentina; mceciliacastilla@gmail.com; 7Instituto de Investigaciones de Biodiversidad Argentina (PIDBA), Facultad de Ciencias Naturales e IML, UNT y Fundación Miguel Lillo, San Miguel de Tucumán 4000, Argentina

**Keywords:** rodent perception, public attitudes, rodent management, Chile

## Abstract

This study developed and tested a new tool called the Scale of Attitudes towards Rodents (SARod), which helps measure how people in Chile feel about rodents. Rodents, although important in nature, are often viewed negatively. The survey included 22 questions asking participants about their thoughts and feelings on rodents, focusing on four areas: scientific views, positive feelings, negative emotions, and negative behaviors towards rodents. The study involved 497 people and found that individuals with higher education and more experience with rodents tended to have more positive attitudes. On the other hand, those who supported eliminating rodents had more negative attitudes. Interestingly, there were no major differences between how men and women viewed rodents. This scale is reliable and could be useful in understanding how people from different cultures view rodents. The results can help guide education efforts and policies to promote peaceful coexistence with rodents, improving both public perception and rodent management strategies. Future studies should use this tool in other cultures to see if attitudes vary worldwide.

## 1. Introduction

Despite the order Rodentia’s diversity, encompassing species as disparate as squirrels and capybaras, it is often narrowly perceived as synonymous with rats and field mice. The number of species belonging to the Rodentia order is under continuous review worldwide due to its taxonomic complexity. It is the most diverse order within mammals due to its limited dispersion and ecological specialization, contributing a large amount of endemism [1,2]. In Chile, Rodentia is the most species-rich mammalian order, accounting for 67 of the country’s 163 living mammal species, distributed across 7 families (*Caviidae*, *Chinchillidae*, *Abrocomidae*, *Ctenomyidae*, *Echimyidae*, *Octodontidae*, and *Cricetidae*) and 30 genera. The Chilean rodent fauna includes iconic species such as chinchillas, coypu, and mice. There are also five introduced species—*Castor canadensis*, *Ondatra zibethicus*, *Mus musculus*, *Rattus exulans*, and *Rattus norvegicus*—several of which raise significant sanitary and domestic concerns due to their association with human activity [2].

In urban and rural settings, exotic species like *Rattus norvegicus*, *Rattus rattus*, and *Mus musculus* are dominant, thriving in areas of high human presence where food and shelter are readily accessible. In contrast, native species, particularly those of the genus *Abrothrix*, prevail in natural environments, being adapted to habitats less impacted by human activity where exotic fauna is not dominant [3,4]. The composition of rodent assemblages in Chile varies significantly throughout the country depending on the different habitats (urban, rural, and natural) and environmental gradients (geographical and climatic). The most diverse areas are represented by the mid-latitudes, around 30° and 40°. The presence of temperate forests, such as the Valdivian, also represents areas of high diversity. However, there is an increase in endemism in island and desert areas [1,5,6]. This environmental heterogeneity has contributed to the high endemism of Chilean rodents, with 14 species being exclusively found within the country’s borders [2].

Rodents provide valuable ecosystem services, including pest control, soil health enhancement, and nutrient cycling. They also play a crucial role in seed dispersal and the facilitation of fungal growth [7,8]. They also occupy a pivotal position in food webs, supporting the populations of carnivores, birds of prey, and other predators. Furthermore, they serve as valuable bioindicators, reflecting the health and integrity of ecosystems [9]. Despite their ecological significance, rodents are subject to numerous conservation threats, including habitat loss, the invasion of non-native species, and human–wildlife conflicts. As a result, the conservation status of Chilean rodents varies greatly, with island populations facing particularly dire threats, including extinction risk for certain species [10]. Moreover, recent discoveries of new species and taxonomic revisions underscore the urgent need for further research and targeted conservation efforts to safeguard Chilean rodent diversity [4].

Rodents pose significant threats to human health and welfare, acting as vectors for diseases and causing substantial economic damage through agricultural depredation. Urban rodents, particularly the house mouse, brown rat, and black rat, are among the most problematic pests globally [10]. These rodents serve as reservoirs for a diverse array of pathogens, including viruses (e.g., *Seoul hantavirus*), bacteria (e.g., *Leptospira* spp.), protozoa (e.g., *Toxoplasma gondii*), and helminths (e.g., Hymenolepis diminuta) [11,12]. There are numerous studies and documents devoted to the development of management and conservation strategies for rodents throughout the world [13]. The tendency is to reduce the use of chemical products, to maintain natural controls (natural enemies), and to protect threatened species or those with ecological functions important for human well-being. Public attitudes towards rodents are pivotal for their conservation and responsible management [14,15]. Regarding negatively perceived animals, individuals’ past experiences, beliefs, perceptions, and behavioral intentions can significantly influence their support for conservation initiatives, pest control measures, and ethical interactions [16,17,18,19,20,21], as in the case of rodents.

Previous research investigating public attitudes towards rodents is relatively limited, particularly within the Latin American region, where Fitte et al.’s [22] study in La Plata, Argentina, stands out as a notable exception. This study examined social perceptions of urban rodents and their associated health risks in two contrasting neighborhoods, revealing a shared concern about rodent-borne diseases and emphasizing the need for public-centered pest management strategies. While both neighborhoods perceived rodents as a threat, El Mondongo demonstrated greater knowledge of rodent species and habitats. In Malawi, Donga et al. [23] found that farmers widely recognized rodents as agricultural pests but often lacked awareness of the associated health risks. A media campaign in the Philippines [24] successfully improved farmers’ knowledge, attitudes, and practices towards environmentally friendly rodent management, leading to increased crop yields and more selective pest control. Studies in Tanzania and Ethiopia [25] revealed that while farmers acknowledged the detrimental impact of rodents on agriculture, particularly in monocultures, they often lacked the necessary information for effective management, resulting in inconsistent and ineffective control measures. Morzillo and Mertig [26] analyzed the relationship between demographic and socioeconomic variables and attitudes towards rodents in various urban contexts. Their findings revealed significant differences in attitudes, with males generally expressing more negative views than females and younger, more educated individuals tending to have more positive perceptions.

To summarize, the existing literature emphasizes that public perceptions and management strategies regarding rodents are predominantly centered on their control as pests, particularly within agricultural contexts [23,24,25]. However, three key research gaps emerge from this body of work. As in the order *Chiroptera* [27], in rodents it seems that the positive ecological roles they play, such as seed dispersal or their role as integral parts of the food web, are not sufficiently known. Although some studies suggest that rodenticides and other control measures can negatively impact non-target species, there has been limited exploration of how to mitigate these effects, the significance of protecting beneficial rodent species, or how the public perceives rodents that do not currently pose a direct threat to human health or livelihoods. Second, while public perceptions of rodents have been studied in various regions, including Argentina [22], the United States [26], the United Kingdom [28], the Philippines [23,24], Malawi [22], and Tanzania [25], there is a notable gap in understanding how these perceptions and management practices differ across diverse cultural and geographic settings. This underscores the need for more research that examines regional variations in attitudes toward rodents and their control. Finally, the variability in the questionnaires and scales used to assess public perceptions of rodents and pest management strategies presents a significant challenge. The lack of standardization across these tools makes it difficult to compare results across different studies and regions. While some studies have employed techniques such as factor analysis and Cronbach’s alpha to evaluate the internal consistency of their instruments [25], research on the psychometric properties of these tools remains limited. There is currently no widely accepted, psychometrically sound instrument available for regional or global use, which complicates cross-cultural comparisons of public attitudes toward rodents [29]. Thus, while numerous tools exist in the literature to assess public perceptions of rodents and their management, there is a clear need for the development of a standardized instrument with robust psychometric properties, suitable for use across diverse community populations and cultural contexts. Our research seeks to address this critical gap by contributing to the creation of such a tool, advancing the study of public attitudes toward rodents and their management.

In response to the lack of a psychometrically sound instrument tailored to the Chilean sociocultural context for assessing community attitudes toward rodents, our primary objective was to develop the Scale of Attitudes toward Rodents (SARod) and evaluate its psychometric properties within the Chilean population.

We are aware that attitudes toward rodents are likely to vary significantly depending on the type of rodent, with certain species, such as house mice or native species, potentially eliciting distinct responses. However, as a necessary first step, constructing and validating a multidimensional instrument like the SARod is essential for capturing general attitudes without introducing pre-defined distinctions among rodent types. This common questionnaire, with evidence of validity, will serve as a foundation for future research, enabling the precise comparison of attitudes toward different rodent types across varied cultural and ecological contexts. To this end, after defining the construct and developing an initial pool of items, we set the following specific objectives: (1) to conduct a descriptive analysis of the SARod items; (2) to provide evidence of validity based on the internal structure of the scale; (3) to assess the reliability of the SARod through internal consistency measures; and (4) to examine the validity of the SARod in relation to sociodemographic variables (gender, age, and education), level of interaction with rodents, and willingness to engage in rodent extermination behaviors.

Aligned with our specific objectives and informed by the existing literature, we formulated the following expectations about the instrument’s desirable properties (DP): we anticipated that the SARod would yield a parsimonious scale, with items demonstrating sufficient discriminatory power (DP1) and a multidimensional internal structure (DP2). Additionally, we expected the internal consistency of the factor scores to meet or exceed a threshold of 0.7 (DP3). Finally, we hypothesized that, with an intermediate-to-large effect size, male (H4a), older (H4b), and less-educated (H4c) individuals would exhibit more negative attitudes toward rodents compared to female, younger, and more highly educated participants. Moreover, we expected participants with less frequent interaction with rodents (H4d) and a greater inclination toward extermination behaviors (H4e) to display more negative attitudes than those who had more interaction with rodents and were less disposed to extermination behaviors.

## 2. Materials and Methods

### 2.1. Design

This work presents an instrumental design [30,31]. We consider the methodological recommendations of Abad et al. [32] and Lloret-Segura et al. [33] for decision-making regarding the selection of evidence of validity, reliability, and statistical analysis.

### 2.2. Participants

By means of convenience sampling, we obtained an initial sample of 541 participants. After eliminating cases due to the presence of missing values, the final sample was 497 participants. The mean age was 36.6 years (SD = 12.48), close to the mean population age in the country of 35.8 years (National Institute of Statistics, 2018). Most of the sample identified as female (*n* = 291, 58.6%), had university or higher education (*n* = 394, 79.3%), lived in the central macrozone of the country (*n* = 313, 63.9%), and lived in an urban area (*n* = 424, 85.3%). In addition, 97.2% (*n* = 483) said they have Chilean nationality, and 4.6% (*n* = 23) identified with the Mapuche indigenous community. Table 1 shows descriptive data on sociodemographic variables for the total sample and stratified by subsamples 1 and 2.

### 2.3. Instruments

Ad hoc sociodemographic questionnaire. This includes questions on topics such as gender identification, age, educational level, nationality, identification with native people, city of residence, or income level.

The Scale of Attitudes towards Rodents (SARod) was designed to assess attitudes toward rodents, defined as relatively stable, favorable or unfavorable evaluations of a cognitive, emotional, or behavioral nature [33]. The development of the theoretical dimensions and items was based on Kellert’s [14,15] typology and the existing research on the attitudes of the Chilean community towards endangered species [20,21]. Kellert’s [14,15] typology of attitudes includes nine basic attitudes towards wildlife and their natural habitats. In addition, previous research points to the importance of considering the myths dimension in Chile [20,21]. On the other hand, we also used the “Tripartite Model of Attitudes” [34] to develop the items. This classic model of social psychology defines attitudes as a combination of three main components: cognitive, affective and behavioral. The same attitude can be manifested through these three components. In this way, these three components are observable manifestations of the unobservable theoretical construct that we wish to measure. In this case, the dimensions of attitudes towards rodents. Each of these items was designed/selected considering its representativeness and relevance to the theoretical construct. Adequacy was also considered, i.e., the inclusion of a sufficient number of items to represent the key aspects of the constructs, while avoiding irrelevant or redundant items [31]. This may involve mixing (or not) components to represent each dimension.

In designing the SARod, participants were prompted to consider rodents in a general sense, without focusing on specific species. This approach was taken to ensure that the scale captures broad attitudes toward rodents, facilitating an unbiased measure that can be applied across various contexts. By capturing general attitudes, the SARod aimed to provide a foundation for future studies that may wish to investigate attitudes toward particular rodent species or types. We initially developed 32 items, structured into four theoretical dimensions: (1) Scientistic, defined as “primary interest in the physical attributes and biological functioning” of rodents [12] (p. 179), with 6 items; (2) Positivistic, focusing on ecological concern for rodents, their habitat, and their practical value, with 7 items (this dimension merges Kellert’s ecologistic and utilitarian attitude types); (3) Negativistic, centered on “active avoidance of (rodents) due to dislike or fear” [14] (p. 180), with 15 items; and (4) Myths, referring to “beliefs, legends, or non-scientific knowledge” about rodents [21] (p. 4), with 4 items. A 5-point Likert scale was used, ranging from 1 (“Completely disagree”) to 5 (“Completely agree”). The psychometric properties of the SARod are detailed in the Section 3.

Disposition toward ad hoc rodent extermination behaviors was measured as a quantitative variable assessing agreement with extermination behaviors. A factorial index was created using three items excluded from the initial SARod (“Rodents should be exterminated”; “We should preventively eliminate rodent burrows”; and “Rodenticides should be used for rodent control”). The corrected item-total correlations ranged from 0.606 to 0.719, with good internal consistency (Ω = 0.835). This index showed no significant skewness or kurtosis and explained 55.55% of the variance. Higher scores indicate greater agreement with extermination behaviors.

### 2.4. Procedure

A cultural adaptation and validation of various questionnaires was conducted through a feedback process for each section of the survey [14,15,16,17,18,20]. This process enabled the adaptation of the questionnaire to the linguistic and cultural context of Chile. The resulting initial version consisted of 34 items distributed across six dimensions: 7 for “Scientism”, 7 for “Positivism”, 4 for “Negative Emotions”, 5 for “Negative Behavior”, 7 for “Negative Cognition”, and 4 for “Myths and Culture”. The questionnaire was then submitted for expert evaluation by individuals with master’s or doctoral degrees, and reviewed by specialists in ecology, conservation, sociology, and psychology. Each expert assessed the relevance of each item and its alignment with the intended dimension, as well as the wording of the instructions and the phrasing of each question. Based on expert feedback and analysis of the responses, some items were revised, and certain questions were either eliminated or adapted to better fit the dimensions being evaluated. Following this, the survey underwent a pilot evaluation with students from two universities (Santo Tomás and Universidad de la Frontera), where response difficulties were noted, particularly in the “Scientistic” and “Negative Emotion” dimensions. Consequently, one item from each of these dimensions was removed. The final validated version consisted of 32 items, distributed as follows: 6 items for “Scientistic”, 7 for “Positivist”, 3 for “Emotional Negativist”, 5 for “Behavioral Negativist”, 7 for “Cognitive Negativist”, and 4 for “Myths and Culture”.

### 2.5. Data Analysis

First, we used descriptive and frequency statistics to characterize the study sample. Subsequently, to address specific objective 1, we conducted a descriptive analysis of the items, calculating the following statistics: mean, standard deviation, skewness, kurtosis, and corrected item-total correlation for the full sample (CIT-T) and by theoretical dimension (CIT-DT). For the CIT-T calculation, the items from the Negativistic and Myths dimensions were reversed so that, for this analysis, higher scores indicated greater acceptance of the species across all items. Skewness values exceeding ± 2, kurtosis values exceeding ± 7, or corrected item-total correlations below 0.3 were considered indicative of low discriminative ability [31,35].

To address specific objective 2, we conducted a cross-validation process, also referred to as factor replication in new samples, which is essential for establishing the generalizability of the instrument’s factor structure. We randomly split the sample into two subsamples. Using subsample 1, we explored the factor structure of the instrument by first assessing the data’s suitability for exploratory factor analysis (EFA) through Bartlett’s test and the Kaiser–Meyer–Olkin (KMO) measure of sampling adequacy. We also evaluated sample size adequacy, guided by communalities in relation to the number of items per factor. According to Lloret-Segura et al. [33], for sample sizes of 200 participants, communalities between 0.4 and 0.7 and at least 3 or 4 items per factor are considered acceptable. We then conducted the EFA using unweighted least squares as the extraction method and oblique rotation. Items with factor loadings below 0.3 across all factors were eliminated. To further ensure the discriminative ability of items based on the resulting factor structure, we: (1) examined cross-loadings, removing items with loadings greater than 0.3 on two or more factors [35]; (2) analyzed item-total correlations corrected by dimension, removing items with values below 0.3; and (3) assessed bivariate correlations between items within each dimension. Correlations exceeding 0.8 indicated redundancy, while non-significant correlations or those below 0.2 suggested that an item (or items) might not measure the same concept as the rest of the dimension. In both cases, the removal of items was considered.

We then proceeded to replicate the factor structure identified in the EFA using subsample 2 through a confirmatory factor analysis (CFA). We employed the robust unweighted least squares estimator (ULSMV) with a polychoric matrix, given the ordinal nature of the data. Additionally, we examined whether the factors adhered to an oblique model, consisting of four correlated first-order factors (Model 1), or a hierarchical model with four first-order factors and one second-order factor (Model 2). As an alternative to the oblique model, we explored whether the structure permitted cross-loadings between items and latent variables, assessing the fit of the structure using an exploratory structural equation model (ESEM) (Model 3). This approach was considered because psychological variables have been shown to fit better under ESEM, which allows for cross-loadings, compared to the restrictive assumptions of CFA [36]. Finally, as an alternative to the hierarchical model, we evaluated the structure using a bifactor model (Model 4). In this model, a general factor explains the covariation among all items while accounting for specific factors or dimensions simultaneously [37,38].

We assessed the model fit using several indices: the root mean squared error of approximation (RMSEA), comparative fit index (CFI), and Tucker–Lewis index (TLI). A good fit is indicated by CFI and TLI values ≥ 0.95 and RMSEA < 0.05, while an acceptable fit is indicated by CFI and TLI ≥ 0.90 and RMSEA < 0.08. We also used Schwarz’s Bayesian information criterion (BIC) for model comparison, where a lower value suggests better fit.

To address specific objective 3, we evaluated internal consistency using McDonald’s omega coefficient, which is recommended for ordinal data [39]. Lastly, to examine differences in the mean scores of the SARod factors based on sex (male/female), level of interaction with rodents (never/once/more than once), and educational level (undergraduate/university/postgraduate), we conducted comparisons. We also analyzed correlations between the SARod factors, age, and “Willingness to Exterminate Rodents” (specific objective 4). We checked for violations of parametric assumptions using the Kolmogorov–Smirnov and Levene’s tests for the distribution of factor scores across study variables. However, with large sample sizes, these violations are less problematic [40]. For comparing means between two groups, we employed the parametric Student’s *t*-test, using Hedges’ g to calculate the corrected effect size for groups of different sizes. For comparisons involving more than two groups, we used one-factor ANOVA with Scheffe’s post-hoc contrasts, appropriate for groups of unequal sizes. The effect size of the overall ANOVA was calculated using partial eta squared (η^2^p), with values of 0.01, 0.06, and 0.14 indicating small, medium, and large effects, respectively. For Scheffe comparisons, Cohen’s d was used to assess effect size, where 0.2 represents a small effect, 0.5 an intermediate effect, and 0.8 or higher a large effect [41]. Pearson’s correlation was applied to test relationships between quantitative variables. We used SPSS 24 for Windows, Mplus 7, Factor 10.9, and JASP 0.18.3 for the analyses.

## 3. Results

### 3.1. Descriptive Analysis of the Items

Appendix A presents the descriptive statistics for the 32 items. All items show acceptable values for skewness, kurtosis, and corrected item-total correlation, both for the overall scale and within each theoretical dimension. These results indicate good discriminative capacity, and as a result, none of the items were eliminated based on these criteria.

### 3.2. Analysis of the Factorial Structure of the SARod, and Descriptive Analysis of the Items According to the Factorial Structure Obtained

We used subsample 1 (*n* = 248) to explore factor structure. The KMO index of 0.911 and Bartlett’s sphericity test (χ^2^(496) = 4738, *p* < 0.001) indicated that the correlation matrix was suitable for exploratory factor analysis (EFA). With the exception of items 23 and 29, which had communalities of 0.362 and 0.378, respectively, the communalities ranged from 0.413 to 0.787, confirming that the sample size was adequate for conducting the EFA.

The initial analysis resulted in six factors explaining 55.2% of the variance. However, after progressively eliminating items due to factor weights below 0.3 (items 15 and 21), cross-loadings greater than 0.3 on multiple factors (items 1, 2, 17, 19, 24, 30, and 31), and the removal of item 28 based on theoretical considerations—because it lacked a clear connection to the other items in its dimension—it was necessary to perform a new EFA with the remaining items. On this occasion, we obtained a KMO index of 0.904. Bartlett’s sphericity test was significant (χ2_(231)_ = 2948, *p* < 0.001). Both results indicated that the correlation matrix is suitable to be subjected to AFE. With the exception of four items (18, 23, 24 and 29) with communalities between 0.319 and 0.384, the communalities ranged from 0.408 to 0.769. We assumed a sufficient sample size considering these values and their relation to the number of items per factor. As a result, we obtained a four-factor structure explaining 54.75% of the variance: (1) the Scientistic factor remains with four items and factor weights between −0.780 and −0.907; (2) the Positivist factor remains with seven items and factor weights between 0.428 and 0.880; (3) a new factor is configured that brings together three items from the Negativistic theoretical dimension, and one item from the Myths dimension with factorial weights between 0.401 and 0.790, which we have named Emotional Negativistic. This dimension reflects stereotypes and emotions of rejection towards rodents generated by fear or disgust; (4) finally, a last factor is configured with six items from the Negativistic dimension and one from the theoretical dimension of Myths with factorial weights between 0.458 and 0.811. We have named this fourth dimension as Cognitive and Behavioral Negativistic, which manifests cognitions about rodents as a threat to the ecosystem and human beings derived in control behaviors. Table 2 displays the factor loadings of the items within each dimension.

Additional analyses of the discriminative capacity of the items, based on the corrected item-total correlation for each dimension, yielded values above 0.3 in all cases. Furthermore, all bivariate correlations between items within each dimension were significant, with none falling below 0.2. Although items 3 and 4 exhibited a high correlation (0.829), both were retained in the scale, as they met all other discriminative capacity criteria. Eliminating either item would have resulted in a notable reduction in the internal consistency of the respective dimension.

Finally, we evaluated the model fit in subsample 2, using the factor structure derived from the EFA in subsample 1 (See Table 3). Among the tested models, the oblique models provided a better fit, with Model 3 showing the best overall fit. However, we encountered a Heywood case [42] in this model, where parameter 16 exceeded unity within the Emotional Negativistic factor. This required respecification of the original model, setting a minimum value approaching zero [43]. Importantly, this adjustment did not affect the overall fit indices.

Figure 1 illustrates Model 3, displaying the factor loadings of the items in the factors with the highest weights, consistent with the structure derived from the EFA. Additionally, Table 4 provides the factor loadings of all items across the four factors in Model 3, along with the correlations between the factors.

### 3.3. Internal Consistency

All factors demonstrated good to excellent internal consistency in both subsample 1 and subsample 2 (see Table 4).

### 3.4. Evidence of Validity of the SARod Based on the Relationship with Other Variables: Gender, Educational Level, Age, Interaction with Rodents, and Willingness to Exterminate Rodents

No gender differences were identified in any of the dimensions (see Table 5). However, participants with a higher level of interaction with rodents scored higher in the Scientistic and Positivistic dimensions, and lower in the Emotional Negativistic dimension. These significant differences demonstrated a medium to large effect size.

Regarding educational level (see Table 6), significant differences were found between groups for all dimensions except the Emotional Negativistic dimension. However, these differences do not hold within groups when using Scheffe’s contrast for the Scientistic and Cognitive and Behavioral Negativistic dimensions. For the Positivistic dimension, participants with lower education scored significantly lower than those with undergraduate or graduate education. The effect size for the overall ANOVA test on this dimension is small, as indicated by the partial η^2^ value. However, post-hoc tests revealed small to medium effect sizes according to Cohen’s d (d = 0.432 and d = 0.366, respectively).

Finally, as shown in Table 7, age and disposition toward exterminating rodents correlate significantly and positively with the Scientistic and Positivistic dimensions, and significantly and negatively with the Emotional Negativistic dimension. Additionally, the variable “disposition to extermination in rodents” also correlates significantly and negatively with the Cognitive and Behavioral Negativistic dimension.

## 4. Discussion

Given the potential threat posed by negative attitudes toward rodent conservation and the lack of a psychometrically sound instrument to measure these attitudes in Chile, we developed the Scale of Attitudes toward Rodents (SARod). The SARod consists of 22 Likert-type items with demonstrated discriminative capacity, structured into four correlated factors identified through exploratory structural equation modeling (ESEM): Scientistic, Positivistic, Emotional Negativistic, and Cognitive and Behavioral Negativistic. The analyses conducted on this four-factor structure provide strong evidence of internal validity, supporting the SARod’s applicability in the Chilean population. The internal consistency of the factor scores, demonstrated in this study, provides further evidence of the scale’s reliability. Additionally, the relationships observed between SARod factors and relevant sociodemographic and behavioral variables, as discussed in the literature, offer further validity evidence based on associations with external variables. These findings collectively suggest that the SARod is a robust tool for assessing attitudes toward rodents, with potential implications for both conservation efforts and public health strategies in Chile.

First, and in accordance with desirable property 1, we observed that all items demonstrated discriminative ability. Specifically, each item effectively differentiated participants’ attitudes towards rodents, as they did not exhibit significant skewness or kurtosis violations or high corrected item-total correlation values. This confirms the suitability of the final 22 items for inclusion in the SARod. Below, we review the most salient descriptive aspects of the items, organized according to the resulting four-factor structure.

The Scientistic factor reflects a more rational appreciation of rodents, emphasizing the interest in acquiring accurate knowledge about the species. The mean scores of participants on each item in this factor suggest a higher agreement with statements that involve less personal investment. In fact, item 6, “I would like to read scientific publications or other material about rodents”, received the highest agreement among participants, while item 5, “I would like to be able to teach on rodent-related topics”, elicited the lowest agreement. The Positivistic factor represents a favorable perception of rodents, focusing on their ecological importance within the ecosystem. Notably, item 10, “Having rodents in the vicinity of where I live can be positive or beneficial”, generated the least agreement among participants, with a mean score significantly lower than the other items. We interpret this as indicating that, although participants recognize the positive contributions of rodents, accepting them in close proximity to their living spaces is more challenging.

The Emotional Negativistic factor groups items related to fear and disgust toward rodents, consistent with the literature that identifies negative emotional responses as key components in attitudes toward animals regarded as pests [25]. The high factor loading of item 16, “Rodents make me afraid”, in the confirmatory factor analysis (CFA), despite an intended lower factor weight, underscores the significance of emotional responses in the perception of rodents, which aligns with findings from previous cross-cultural studies [24]. Notably, item 29, “Rodents are a symbol of bad omens”, exhibited a lower mean score compared to other items within this factor. This item was initially assigned to the eliminated theoretical dimension of Myths. This result suggests that while emotions eliciting the highest levels of agreement may be influenced by cultural beliefs, the mystical aspect of such beliefs appears to have less impact on the emotions evoked by rodents. The Cognitive and Behavioral Negativistic factor includes perceptions of rodents as a threat and associated control behaviors. Studies by Stuart et al. [29] and Donga et al. [23] similarly emphasize the prevalence of negative attitudes toward rodents and the widespread implementation of control practices due to the economic and health risks they pose. Although the range of mean values for this factor is relatively narrow, it is noteworthy that item 20, “Rodent reproduction should be controlled or stopped”, received the lowest level of agreement among participants. While another item in this factor, “I think we should take measures to control rodents” (item 18), also addresses the need for intervention, item 20 is the only one that explicitly calls for direct interference. This finding is consistent with the results of Fitte et al. [22], who found that approximately half of participants in an Argentine sample were willing to improve home cleanliness to reduce rodent presence, but willingness to adopt specific control measures, such as the use of poison and traps, decreased to 20% and 10%, respectively.

Second, we have defined desirable property 2 as that the SARod would have a multifactorial structure, and the results support this proposition. The four-factor correlated structure of the SARod aligns with previous literature on attitudes toward animals, which suggests that attitudes are multifaceted and encompass cognitive, emotional, and behavioral components [13,14,15,16,17]. The Scientistic and Positivistic dimensions show a significant and positive correlation with a high value, indicating that participants who hold a positive view of rodents also tend to have a more rational appreciation of the species, and vice versa. This is consistent with earlier studies that suggest individuals with greater scientific knowledge tend to exhibit more positive attitudes towards wildlife [16,17,18,19,21]. Similarly, the Emotional Negativistic and Cognitive and Behavioral Negativistic dimensions are also significantly and positively correlated, suggesting that individuals who harbor negative beliefs about rodents and express intentions to engage in control behaviors are more likely to experience negative emotions toward the species, and vice versa. This relationship reflects findings from previous research that indicate these types of attitudes are often correlated [19,21], in line with the tripartite model of attitudes [33]. Furthermore, these negative factors are significantly and negatively associated with the Scientistic and Positivistic factors. This suggests that individuals with a scientific and ecological interest in rodents tend to have fewer negative attitudes on emotional, cognitive, and behavioral levels; conversely, those with stronger negative emotions and beliefs about rodents are less likely to appreciate the species from a scientific or ecological perspective.

Although Model 3 (i.e., ESEM) achieved better fit indices, all the structures subjected to CFA analysis demonstrated adequate values. This finding has important implications for both result interpretation and the practical application of this tool. On one hand, the superior fit of the ESEM model compared to the oblique model (Model 1) suggests that the psychological constructs being measured do not function in an entirely distinct manner. While individual items are more influential in defining a specific construct, they also contribute to the configuration of the other three constructs [36]. On the other hand, the adequate fit of Model 2 (hierarchical) confirms that, despite the superiority of the ESEM model, it is possible to consider the existence of a higher-order factor, suggesting a general factor representing attitudes toward rodents. Furthermore, the adequate fit of the bifactor model (Model 4) indicates that there is indeed a general factor accounting for the covariation among all items. This supports the appropriateness of considering a global score for the scale, in addition to the individual dimensions [37,38].

In our desirable property, we proposed achieving internal consistency of the factor scores equal to or greater than 0.7. The coefficients obtained in subsamples 1 and 2 range between 0.807 and 0.945. This result ensures the measure’s accuracy in reflecting participants’ true scores on the theoretical construct based on their observed test scores. Moreover, the adequate internal consistency of the scores further indicates the reliability and replicability of the measure. In other words, if the measure were applied again to the same or a similar population, we would expect to obtain comparable results, even with variations in conditions such as the method of administration, the time interval between measurements, or the person administering the test [31,32].

We also proposed several hypotheses regarding the relationship between SARod factors and sociodemographic and behavioral variables. Our analysis reveals that younger participants exhibit more scientistic and positivistic attitudes, and less emotional negativism toward rodents. Additionally, educational level significantly influences attitudes, with participants who have higher education viewing rodents more positively. This pattern aligns with findings by Morzillo and Mertig [26], who reported that individuals with higher education tend to have more positive perceptions of urban wildlife. Our results also reflect trends observed by Boso et al. [20] and Pérez et al. [21] in studies on attitudes toward bats in Chile, another species often associated with negative perceptions. Thus, we confirm the validity of hypotheses 4b and 4c. However, unlike Morzillo and Mertig [26], who found more negative perceptions of rodents among men, our study did not reveal statistically significant differences in SARod factors based on participant gender. Consequently, hypothesis 4a is not supported. This discrepancy may be attributed to cultural and contextual differences between the samples, as Morzillo and Mertig [26] studied an urban American population, while our sample consists of a Chilean population that is not exclusively urban. Our findings also differ from those of Boso et al. [20] and Pérez et al. [21], where men exhibited greater scientific interest and less negative attitudes toward bats. These variations highlight the importance of examining cross-cultural differences in rodent perceptions, particularly in relation to gender norms and the social representations of species often unfairly perceived as threats to humans and ecosystems.

Finally, we examined the relationship between SARod factors and behavioral variables, confirming hypotheses 4d and 4e. Our results indicate that frequent interaction with rodents is associated with more positive attitudes and fewer negative emotional responses toward these animals. This finding aligns with the study by Flor and Singleton [24], which demonstrated that familiarity and knowledge can enhance attitudes and improve handling practices. Similarly, Boso et al. [20] found that attitudes toward bats were more positive when participants had previously encountered them. Moreover, participants with a stronger disposition toward rodent extermination exhibited more negative attitudes across all dimensions. This result is particularly significant for its implications. On one hand, it underscores the potential of educational campaigns aimed at mitigating negative attitudes as a strategy to reduce extermination behaviors. On the other hand, it highlights the SARod as a valuable tool for identifying individuals not only with negative attitudes toward rodents but also those with a higher likelihood of engaging in extermination behaviors. This functionality could be instrumental in applying educational interventions in a more targeted and effective manner within such populations.

The Scale of Attitudes towards Rodents (SARod) represents a significant advance in the measurement of attitudes towards rodents. The development of this instrument provides a detailed and specific assessment of attitudes towards rodents in the Chilean context, albeit subject to the particularities of the sample, and establishes a framework that can be adapted and applied to other regions and cultures. This is particularly relevant in a global context where wildlife management practices and cultural perceptions of rodents vary widely. The transcultural development of this tool could facilitate the comparison of attitudes across different populations [30], an essential step in the development of culturally sensitive and effective rodent management strategies. By incorporating dimensions that assess cognitive, emotional, and behavioral components, the SARod offers a comprehensive understanding of human attitudes toward rodents, which can inform conservation policies and environmental education programs. Ultimately, the SARod has the potential to contribute to the harmonization of wildlife conservation and management efforts worldwide by enabling a more accurate and comparable assessment of public attitudes toward rodents.

The insights gleaned from the SARod scale have tangible implications for various development sectors. In the realm of food management, understanding public attitudes towards rodents can facilitate the development of more effective awareness campaigns that promote preventive measures among agricultural workers and local communities. This could lead to reduced crop losses without the need for harmful rodenticides. Within public health, the scale can identify population segments with heightened fear or misconceptions about rodents, enabling targeted health education that addresses zoonotic disease risks and promotes safe coexistence practices. Moreover, in the field of pest control regulation, recognizing the social perceptions of rodents can inform policy-making and regulatory approaches that balance ecological concerns with human health priorities. By integrating the findings of the SARod scale, these sectors can develop strategies that are culturally sensitive, reduce reliance on aggressive control methods, and foster a more sustainable approach to rodent population management.

This study has several limitations that should be considered when interpreting the results. First, the use of convenience sampling introduces a level of bias that restricts the generalizability of the findings, as the sample may not fully represent the broader Chilean population. For instance, in our case, nearly 80% of the sample has higher education, compared to the regional average of 27.3%. Similarly, the sample in this study includes a higher proportion of individuals with high socioeconomic status (14%) than the national average (9.8%) [44]. Additionally, while the Scale of Attitudes towards Rodents (SARod) demonstrated strong psychometric properties, these properties should be tested in more diverse samples and across different regions to confirm the consistency of the results. Furthermore, although sociodemographic variables such as gender and educational level were considered, other contextual and cultural factors that may influence attitudes towards rodents—such as religious beliefs or local traditions—were not explored in depth. Finally, another significant limitation of this study is the absence of an analysis of the spatial and temporal dynamics of rodent populations, which can exhibit considerable variation across regions and seasons. These fluctuations may influence public attitudes and perceptions towards rodents, as communities experiencing seasonal surges in rodent populations or frequent human-rodent interactions might demonstrate heightened negative attitudes. This factor could potentially limit the generalizability of our findings to different contexts. Future research employing the SARod scale should account for such temporal and spatial variations to gain a more comprehensive understanding of how fluctuating rodent populations impact public attitudes over time and across diverse geographic areas.

## 5. Conclusions

This study developed and examined the psychometric properties of the Scale of Attitudes towards Rodents (SARod), a tool designed to assess public attitudes toward rodents in the Chilean context. The results revealed a robust factor structure comprising four dimensions: Scientistic, Positivistic, Emotional Negativistic, and Cognitive and Behavioral Negativistic. While the ESEM model provided the best fit, the adequate fit of the bifactor model supports the use of an overall scale score. Additionally, the factor scores demonstrated high internal consistency, and the SARod showed validity in relation to sociodemographic and behavioral variables. The findings underscore the complexity of attitudes toward rodents, influenced by factors such as age, educational level, and prior interaction with these animals. More positive attitudes were linked to higher educational levels, younger age, and greater familiarity with rodents, whereas a greater disposition toward rodent extermination was associated with more negative attitudes.

Similar to the trajectories followed by other conservation programs (e.g., Latin American and Caribbean Network for the Conservation of Bats, RELCOM), which represent animals with varying degrees of conservation threat and conflicts with humans, rodents must be considered as an order and each species in particular. One response is to create programs or actions, adapted to the culture and legislation of each country, that contemplate research, conservation, and management actions, and to engage in education and dissemination in order to reduce negative human attitudes towards rodents based on the ecosystem services they provide [27]. This requires a coexistence approach that contemplates both positive and negative interactions, and comprehensive planning at different sociocultural and spatial scales [45]. Actions associated with damage reduction (avoiding or compensating economic losses, preventing zoonoses, etc.) and monitoring population abundance have a great impact [46].

## Figures and Tables

**Figure 1 animals-14-03239-f001:**
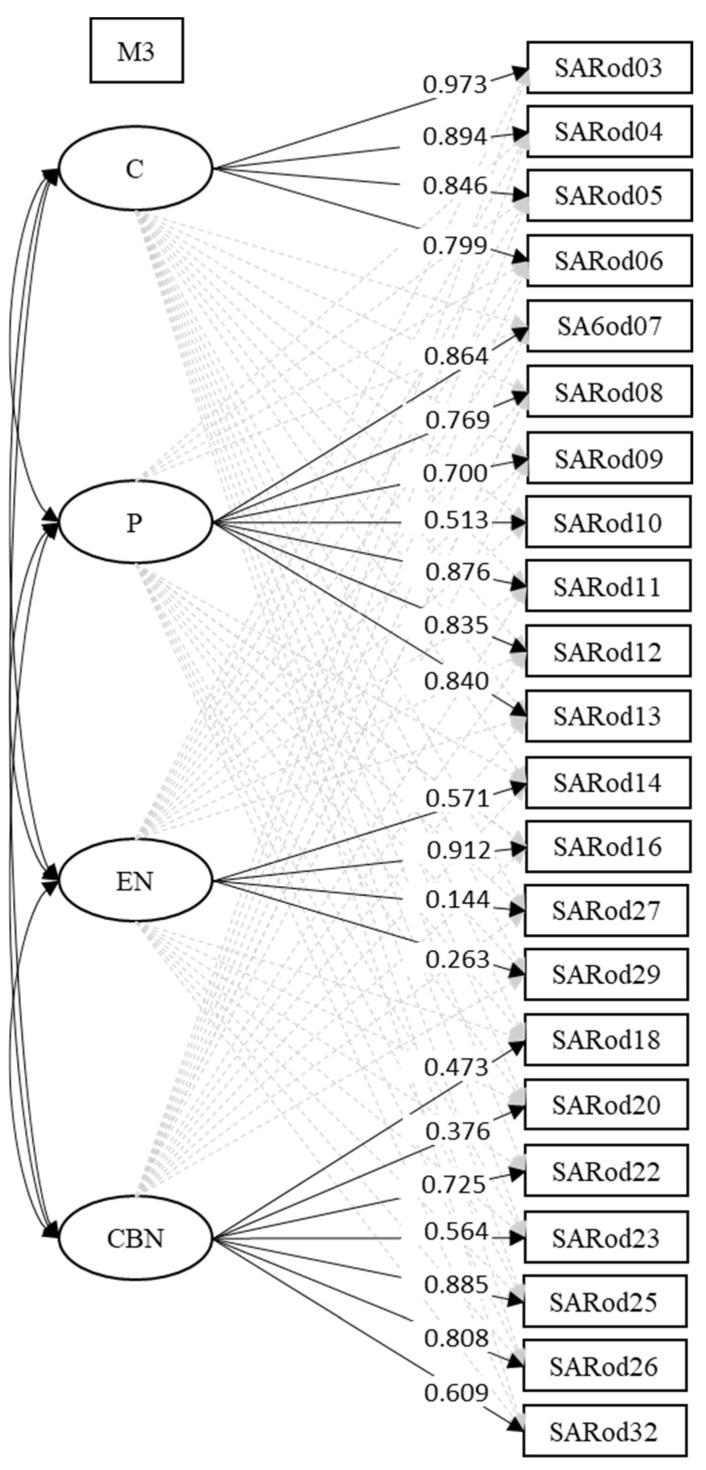
ESEM Model representation corresponding to Model 3.

**Table 1 animals-14-03239-t001:** Descriptive data of the total and stratified sample by subsamples.

			Sub-Sample	
		Total (*n* = 497)	Sub-Sample 1 (*n* = 248; 49.9%)	Sub-Sample 2 (*n* = 249; 50.1%)
Variables	Values	*n* (%)	*n* (%)	*n* (%)
Gender	Male	200 (40.2)	106 (42.7)	94 (37.8)
	Female	291 (58.6)	138 (55.6)	153 (61.4)
	Non-binary	5 (1)	4 (1.6)	1 (0.4)
Chilean nationality	No	14 (2.8)	7 (2.8)	7 (2.8)
	Yes	483 (97.2)	241 (97.2)	242 (97.2)
Identification with native people	No	470 (94.6)	235 (98.8)	235 (94.4)
	Mapuche	23 (4.6)	10 (4)	13 (5.2)
	Other	4 (0.8)	3 (1.2)	1 (0.4)
Geographic macrozone	North	19 (3.9)	11 (4.5)	8 (3.3)
	Center	313 (63.9)	151 (61.6)	162 (66.1)
	South	158 (32.2)	83 (33.9)	75 (30.6)
Income level	Low	156 (31.4)	79 (31.8)	77 (31)
	Middle-low	122 (24.5)	61 (24.6)	61 (24.5)
	Middle	151 (30.4)	71 (28.6)	80 (32.1)
	High	68 (13.7)	37 (14.9)	31 (12.4)
Educational Level	Undergraduate	103 (20.7)	50 (20.2)	53 (21.3)
	University	248 (49.9)	128 (51.6)	120 (48.2)
	Postgraduate	146 (29.4)	70 (28.2)	76 (30.5)
Sector	Rural	73 (14.7)	44 (17.7)	29 (11.6)
	Urban	424 (85.3)	204 (82.3)	220 (88.4)

**Table 2 animals-14-03239-t002:** Factor weights of the items in each of the factors resulting from the AFE in subsample 1, and corrected item-total correlation by dimension.

Items		Factors				CITC-D
		1	2	3	4	
Scientistic						
3	I would like to be part of congresses, seminars, or scientific activities that involve learning about rodents.	**−0.907**	0.006	0.018	−0.017	0.861
4	I would like to exchange knowledge with other people about rodents.	**−0.906**	−0.025	−0.024	0.033	0.849
5	I would like to be able to teach rodent related subjects.	**−0.865**	−0.055	−0.052	−0.001	0.813
6	I would like to read scientific publications or other material about rodents.	**−0.780**	0.129	0.093	0.004	0.767
Positivistic						
7	I believe that rodents are important for the functioning of ecosystems.	0.026	**0.880**	0.018	0.068	0.763
8	I believe that rodents should be protected by people.	−0.081	**0.628**	−0.141	−0.120	0.752
9	We must learn to coexist with rodents.	−0.056	**0.559**	−0.164	−0.035	0.663
10	Having rodents in the vicinity of where I live can be positive or beneficial for me.	−0.110	**0.428**	−0.245	−0.103	0.608
11	I consider that rodents have an importance associated with seed dispersal.	0.012	**0.757**	−0.125	0.027	0.740
12	Rodents are key players in food chains	−0.064	**0.832**	0.046	0.030	0.753
13	Rodents imply an increase in biodiversity in protected areas	−0.072	**0.777**	0.084	−0.027	0.722
Emotional Negativistic						
14	I find rodents unpleasant	0.225	−0.111	**0.571**	0.126	0.64
16	Rodents scare me	0.069	−0.031	**0.790**	−0.024	0.697
27	Rodents are aggressive	−0.036	−0.096	**0.409**	0.205	0.467
29	Rodents are a symbol of bad omens	0.020	−0.180	**0.401**	0.018	0.454
Cognitive and Behavioral Negativistic						
18	I believe that we should take measures to control rodents.	−0.051	−0.116	0.159	**0.485**	0.538
20	Rodent reproduction must be controlled or stopped.	0.043	−0.280	0.044	**0.458**	0.509
22	Rodents cause damage to agricultural crops	−0.032	−0.085	−0.206	**0.811**	0.604
23	Rodents eliminate or compete with other species of wild animals	0.028	−0.005	−0.050	**0.572**	0.471
25	Rodents can contaminate my water sources.	0.093	0.154	0.156	**0.600**	0.580
26	Rodents cause damage to machinery, breakage of cables or hoses.	−0.034	0.163	0.158	**0.543**	0.485
32	In the presence of rodents, I could get sick.	0.097	−0.005	0.185	**0.457**	0.524

Note: In bold type are the factorial weights of the items in the factors to which they are finally assigned.

**Table 3 animals-14-03239-t003:** Fit index of models subjected to AFC.

Models	χ^2^	gl	CFI	RMSEA	TLI	BIC
M1. Oblique: 4 first-order correlated factors	402.777	203	0.953	0.063 (0.054–0.072)	0.947	4498.070
M2. Hierarchical: 4 first-order factors and one general factor	478.056	210	0.937	0.072 (0.063–0.080)	0.931	4498.070
M3. Oblique: 4 first-order factors (ESEM)	245.266	149	0.977	0.051 (0.039–0.062)	0.965	4490.070
M4. Bifactor: 4 first-order factors and a general factor.	401.641	192	0.951	0.066 (0.057–0.075)	0.941	4498.070

Note: χ^2^ = chi square; gl = degrees of freedom; CFI = comparative fit index; RMSEA = root mean square error of approximation; TLI = Tucker–Lewis index; BCI = Bayesian information criterion.

**Table 4 animals-14-03239-t004:** Factor weights of the items in the factors according to Model 3; Correlations between factors in full sample and McDonald omega coefficient per dimension and in each subsample.

	Factors					
	Scientistic	Positivistic	Emotional Negativistic		Cognitive and Behavioral Negativistic	
3	**0.973**	−0.026	0.044		−0.001	
4	**0.894**	0.047	−0.012		−0.011	
5	**0.846**	0.033	0.047		0.068	
6	**0.799**	0.111	−0.019		0.026	
7	0.073	**0.864**	0.007		0.018	
8	0.086	**0.769**	0.016		−0.115	
9	0.049	**0.700**	−0.087		−0.164	
10	0.177	**0.513**	0.002		−0.204	
11	0.018	**0.876**	0.059		0.007	
12	0.065	**0.835**	−0.126		0.071	
13	0.058	**0.840**	−0.013		0.056	
14	−0.221	−0.107	**0.571**		0.259	
16	−0.094	−0.043	**0.912**		0.068	
18	0.055	−0.245	0.047		**0.473**	
20	0.078	−0.413	0.144		**0.376**	
22	0.067	−0.104	−0.126		**0.725**	
23	−0.006	0.097	−0.045		**0.564**	
25	−0.045	0.200	0.071		**0.885**	
26	−0.081	0.138	−0.050		**0.808**	
27	0.012	−0.199	**0.144**		0.456	
29	0.028	−0.474	**0.263**		0.042	
32	0.021	−0.092	0.113		**0.609**	
Correlations						
Scientistic	-	0.512 **	−0.402 **		−0.237 **	
Positivistic		-	−0.580 **		−0.437 **	
Emotional Negativistic			-		0.548 **	
Cognitive and Behavioral Negativistic					-	
McDonald Omega Ratio						
Subsample 1	0.945		0.933	0.812		0.829
Subsample 2	0.939		0.945	0.807		0.850

Note: ** = *p* < 0.01.; in bold type are the factorial weights of the items in the factors to which they are finally assigned.

**Table 5 animals-14-03239-t005:** Mean differences in SARod factors according to sex, degree of interaction with rodents.

	Male (*n* = 200)	Female (*n* = 291)				
	Mean (SD)	Mean (SD)	t	df	*p*	*g*
Scientistic	3.14 (1.07)	3.12 (1.07)	0.188	489	0.851	--
Positivistic	3.91 (0.79)	3.91 (0.78)	0.089	489	0.929	--
Emotional Negativistic	2.47 (0.84)	2.49 (0.92)	−0.247	489	0.805	--
Cognitive and Behavioral Negativistic	3.50 (0.64)	3.40 (0.67)	1.621	489	0.106	--
	Interaction with rodents					
	Never or once (*n* = 66)	More than once (*n* = 630)				
Scientistic	2.61 (0.88)	3.24 (10.09)	−4.448	494	<0.001	0.587
Positivistic	3.53 (0.73)	3.99 (0.77)	−4.508	494	<0.001	0.595
Emotional Negativistic	2.91 (0.82)	2.40 (0.88)	4.402	494	<0.001	0.581
Cognitive and Behavioral Negativistic	3.56 (0.55)	3.41 (0.68)	1.693	494	0.091	--

**Table 6 animals-14-03239-t006:** Mean differences in the SARod factors according to educational level.

	Education Level					
	Undergraduate (*n* = 103)	University (*n* = 248)	Postgraduate (*n* = 146)			
	Mean (SD)	Mean (SD)	Mean (SD)	*F*	η^2^_p_	Post-Hoc (Scheffe)
Scientistic	3.01 (1.03)	3.28 (1.01)	3.04 (1.21)	3.236 *		
Positivistic	3.67 (0.826)	4.01 (0.72)	3.95 (0.83)	6.992 **	0.028	1 < 2, 3
Emotional Negativistic	2.59 (1.00)	2.40 (0.84)	2.49 (0.89)	1.737		
Cognitive and Behavioral Negativistic	3.59 (0.66)	3.40 (0.62)	3.38 (0.74)	3.441 *		

Note: 1 = Undergraduate; 2 = University; 3 = Postgraduate; * < 0.05, ** < 0.01.

**Table 7 animals-14-03239-t007:** Pearson correlations. Age and extermination disposition in rodents with SARod dimensions.

	Dimensions SARod			
	Scientistic	Positivistic	Emotional Negativistic	Cognitive and Behavioral Negativistic
Age	−0.129 **	−0.173 **	0.152 **	0.062
DER	−0.296 **	−0.622 **	0.615 **	0.592 **

Note: DER = disposition to extermination in rodents; ** < 0.01.

## Data Availability

The data presented in this study are available on request from the corresponding author. The data are not publicly available due to restrictions on privacy.

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
