# Peer review of "Assessing Rodent Attitudes: The Psychometric Properties of the SARod in a Chilean Context"

_animals, 2024, doi:10.3390/ani14223239_

Round 1
Reviewer 1 Report
Comments and Suggestions for Authors
Comments on Assessing Rodent Attitudes: The Psychometric Properties of the SARod in a Chilean Context
Major comment
This is an interesting and well-written study. The econometrics is quite sophisticated but the theoretical seem weak to me.
I think the analysis and presentation of these results could be much improved if you incorporated some of the thinking from behavioural psychology such as Bagozzi's dual purpose model of behaviour and the idea of cognitive, affective and behavioural engagement. I think this because your factors are mixtures of cognitive engagement, affective engagement and behavioural engagement (as reflected in goal intentions and behavioural intentions).
For example, 3,4,5, and 6 look like behavioural intentions.
7,11,12,13,22,23,25,26 are indicators of cognitive engagement.
14,16,27,29,32 are indicators of affective engagement
8,9,10,18,20 are indicators look like goal intentions (broader than behavioural intentions).
Usually we would predict behavioural engagement (i.e. goal and behavioural intentions) from cognitive and affective engagement (not mix them together into scales). Otherwise, I think you need to provide a much stronger justification as to the selection of statements and why all these statements could be expected to form factors that can be generalised beyond the sample you analyse. Consequently, I think the content of the paragraph commencing at line 516 about the generalisability of the SARod are too strong, especially as you sampled from a population that was (I believe) heavily biased in terms of education.
One option would be, for example, if you (1) factored only on all the original cognitive and affective statements (the myths are a mix of affective and cognitive statements - I think); (2) form scales (separately) for the goal intention and behavioural intention statements; and (3) regress the intention scales on the cognitive and affective scales. I guess you could then form an overall scale by factoring the 4 scales if you wished.
You should also think about reverse scoring negative statements to prevent positive and negative framing biasing the analysis.
Some references:
Bagozzi, R. P. (2006). Consumer Action: Automaticity, Purposiveness and Self-Regulation. Review of Marketing Re-search, 2, 3–42.
Saks AM. 2006. Antecedents and consequences of employee engagement. Journal of managerial psychology. 21(7):600-19.
Samus A, Freeman C, Dickinson KJ, van Heezik Y. 2023. An examination of the factors influencing engagement in gardening practices that support biodiversity using the theory of planned behavior. Biological Conservation. 286:110252.
Shuck B. 2011. Integrative literature review: Four emerging perspectives of employee engagement: An integrative literature review. Human Resource Development Review. 10(3):304-28.
Soane E, Truss C, Alfes K, Shantz A, Rees C, Gatenby M. 2012. Development and application of a new measure of employee engagement: the ISA Engagement Scale. Human resource development international. 15(5):529-47.
Minor corrections.
1. There are a few of small typographical errors in manuscript.
2. Line 136: I struggle with these as hypotheses given you go through an extensive process to ensure the SARod will have these properties. I would prefer they were expressed as desirable properties you were seeking. This would change some parts of the discussion.
3. Line 163: So, the sample was nor representative of the general population in terms of education (in particular) and income? If so, I think you should state this.
Author Response
REVISOR 1
Major comment
This is an interesting and well-written study. The econometrics is quite sophisticated but the theoretical seem weak to me.
I think the analysis and presentation of these results could be much improved if you incorporated some of the thinking from behavioural psychology such as Bagozzi's dual purpose model of behaviour and the idea of cognitive, affective and behavioural engagement. I think this because your factors are mixtures of cognitive engagement, affective engagement and behavioural engagement (as reflected in goal intentions and behavioural intentions).
For example, 3,4,5, and 6 look like behavioural intentions.
7,11,12,13,22,23,25,26 are indicators of cognitive engagement.
14,16,27,29,32 are indicators of affective engagement
8,9,10,18,20 are indicators look like goal intentions (broader than behavioural intentions).
Usually we would predict behavioural engagement (i.e. goal and behavioural intentions) from cognitive and affective engagement (not mix them together into scales). Otherwise, I think you need to provide a much stronger justification as to the selection of statements and why all these statements could be expected to form factors that can be generalised beyond the sample you analyse. Consequently, I think the content of the paragraph commencing at line 516 about the generalisability of the SARod are too strong, especially as you sampled from a population that was (I believe) heavily biased in terms of education.
One option would be, for example, if you (1) factored only on all the original cognitive and affective statements (the myths are a mix of affective and cognitive statements - I think); (2) form scales (separately) for the goal intention and behavioural intention statements; and (3) regress the intention scales on the cognitive and affective scales. I guess you could then form an overall scale by factoring the 4 scales if you wished.
You should also think about reverse scoring negative statements to prevent positive and negative framing biasing the analysis.
Some references:
Bagozzi, R. P. (2006). Consumer Action: Automaticity, Purposiveness and Self-Regulation. Review of Marketing Re-search, 2, 3–42.
Saks AM. 2006. Antecedents and consequences of employee engagement. Journal of managerial psychology. 21(7):600-19.
Samus A, Freeman C, Dickinson KJ, van Heezik Y. 2023. An examination of the factors influencing engagement in gardening practices that support biodiversity using the theory of planned behavior. Biological Conservation. 286:110252.
Shuck B. 2011. Integrative literature review: Four emerging perspectives of employee engagement: An integrative literature review. Human Resource Development Review. 10(3):304-28.
Soane E, Truss C, Alfes K, Shantz A, Rees C, Gatenby M. 2012. Development and application of a new measure of employee engagement: the ISA Engagement Scale. Human resource development international. 15(5):529-47.
A: We welcome your comments and interest in improving this manuscript. We understand your comments from the theoretical perspective mentioned above. However, this paper is based on the Tripartite Concept of Attitudes (Rosenberg and Hovland, 1960). This theory defines attitudes as a combination of three main components: the cognitive component, the affective component, and the behavioural component. This model is widely used in social psychology to understand how people's attitudes towards objects, people, situations or ideas are formed and manifested. Thus, the same attitude can be manifested from these three components. At the same time, these 3 components are observed manifestations of an unobservable theoretical construct (in this case, the attitude towards rodents). From this theoretical perspective, it is coherent to consider all three components at the same level. At the same time, it is not necessary for all three components to be present among the items that make up each dimension. We are looking for a parsimonious instrument and therefore the most representative items and as few as possible.
On the other hand, factorial organisation is a result of the behaviour of the data. There is no prior intention for the items to be grouped in the way they are. Forcing the items into particular factors, so that the structure conforms to theory, means introducing an important bias into the analyses. For this reason, and from a psychometric point of view, we prefer to retain the structure that emerges from the correlation matrix.
In response to your observations, we have tried to further strengthen the theoretical basis of our work by adding this information about the design of the instrument:
The development of the theoretical dimensions and items was based on Kellert's [14,15] typology and the existing research on the attitudes of the Chilean community towards endangered species [20,21]. Kellert's [14,15] typology of attitudes includes nine basic atti-tudes towards wildlife and their natural habitats. In addition, previous research points to the importance of considering the myths dimension in Chile [20, 21]. On the other hand, we also used the "Tripartite Model of Attitudes" [34] to develop the items. This classic model of social psychology defines attitudes as a combination of three main components: cognitive, affective and behavioural. The same attitude can be manifested through these three components. In this way, these three components are observable manifestations of the unobservable theoretical construct that we wish to measure. In this case, the dimen-sions of attitudes towards rodents. Each of these items was designed/selected considering its representativeness and rele-vance to the theoretical construct. Adequacy was also con-sidered, i.e. the inclusion of a sufficient number of items to represent the key aspects of the constructs, while avoiding irrelevant or redundant items [31]. This may involve mixing (or not) components to represent each dimension..
On the other hand, in response to your comment on the paragraph beginning on line 516, we have changed the text:
The Scale of Attitudes towards Rodents (SARod) represents a significant advance in the measurement of attitudes towards rodents. The development of this instrument provides a detailed and specific assessment of attitudes towards rodents in the Chilean context, albeit subject to the particularities of the sample, and establishes a framework that can be adapted and applied to other regions and cultures. This is particularly relevant in a global context where wildlife management practices and cultural perceptions of rodents vary widely. The transcultural development of this tool could facilitate the comparison of attitudes across different populations [29], an essential step in the development of culturally sensitive and effective rodent management strategies.
Minor corrections.
- There are a few of small typographical errors in manuscript.
A: Thank you for pointing this out. We have carefully reviewed the manuscript and corrected all identified typographical errors to improve clarity and readability. We appreciate your attention to detail and the opportunity to enhance the quality of our submission.
- Line 136: I struggle with these as hypotheses given you go through an extensive process to ensure the SARod will have these properties. I would prefer they were expressed as desirable properties you were seeking. This would change some parts of the discussion.
A: Thank you for your comment. We propose hypotheses 1, 2 and 3 as desirable properties. In text:
“Aligned with our specific objectives and informed by the existing literature, we for-mulated the following expectations about the instrument desirable properties (DP): we an-ticipated that the SARod would yield a parsimonious scale, with items demonstrating sufficient discriminatory power (DP1), and a multidimensional internal structure (DP2). Additionally, we expected the internal consistency of the factor scores to meet or exceed a threshold of 0.7 (DP3)”
We made the corresponding modifications in the discussion.
- 3. Line 163: So, the sample was nor representative of the general population in terms of education (in particular) and income? If so, I think you should state this.
A: Thank you for your comment. We added new lines to the limitations of the study in order to state this issue. See the new text below:
“This study has several limitations that should be considered when interpreting the results. First, the use of convenience sampling introduces a level of bias that restricts the generalizability of the findings, as the sample may not fully represent the broader Chilean population. For instance, in our case, nearly 80% of the sample has higher ed-ucation, compared to the regional average of 27.3%. Similarly, the sample in this study includes a higher proportion of individuals with high socioeconomic status (14%) than the national average (9.8%) (Ministerio de Desarrollo Social y Familia, 2021).”
Reviewer 2 Report
Comments and Suggestions for Authors
Congratulations. Very robust tool SARod.
Very few minor comments and/or suggestions.
1) To mention in the backround and discussions section, the relevance of your findings for ecologically-based rodent management as this strategy has been developed in many countries over the world ( see articles from Steve Belmain, Singleton, the Australian Center for Agricultural Research among others)
2) To give some indication of research criteria for targeting most vulnerable population groups to further test the psychometric properties
3) To consider in the study limitation, the variation of spatial and temporal population dynamics of rodents
4) To give, if possible, some concrete exemples of the impact of such findings in the apllication of strategies in different development sectors, such food management, public health, pest control regulations...
Author Response
Congratulations. Very robust tool SARod.
Very few minor comments and/or suggestions.
1) To mention in the backround and discussions section, the relevance of your findings for ecologically-based rodent management as this strategy has been developed in many countries over the world (see articles from Steve Belmain, Singleton, the Australian Center for Agricultural Research among others).
A: Thank you for your comment. We add the following text in back:
There are numerous studies and documents devoted to the development of management and conservation strategies for rodents throughout the world (Brown et al. 2024). The tendency is to reduce the use of chemical products and to maintain natural controls (natural enemies) and to protect threatened species or those with ecological functions important for human well-being.
And this:
“This tool could contribute to the ecological management of rodents”
Also, in response to your fourth comment, you can see that we have added information about this to the discussion.
2) To give some indication of research criteria for targeting most vulnerable population groups to further test the psychometric properties
A: Thank you for highlighting the importance of identifying vulnerable groups for further validation of the SARod. We agree that targeting specific demographic groups, such as individuals in rural areas, low-income populations, or communities frequently exposed to rodent-related risks, is crucial to testing the scale’s robustness across diverse populations. We have added some reflexions in the discussion to outline potential criteria for future research aimed at these groups, which would enhance the scale’s generalizability and practical applicability in different contexts. The new text is as follows:
Furthermore, to further test the SARod’s psychometric properties, future studies should consider targeting population groups that may be more vulnerable to rodent exposure and related risks. Such groups could include residents of rural or low-income areas where rodent interactions are frequent due to environmental or housing conditions. Additionally, occupations that involve regular exposure to rodents, such as agricultural and sanitation workers, represent relevant populations for testing the scale’s reliability and validity. Including diverse demographic samples, such as indigenous communities with unique cultural perceptions of rodents, can also provide insights into the SARod’s cross-cultural applicability. By focusing on these vulnerable groups, future research can strengthen the SARod’s utility in understanding rodent attitudes across various socioeconomic and cultural contexts.
3) To consider in the study limitation, the variation of spatial and temporal population dynamics of rodents.
A: Thank you for this valuable suggestion. We recognize that the spatial and temporal dynamics of rodent populations could influence public attitudes, potentially affecting responses on the SARod scale. We have incorporated this consideration into the limitations section, noting how fluctuations in rodent populations across regions and seasons may impact the generalizability of our findings. This addition underscores the need for future studies to account for these dynamics when applying the SARod in different contexts. We added the following reflection in the limitations paragraph:
Finally, another significant limitation of this study is the absence of an analysis of the spatial and temporal dynamics of rodent populations, which can exhibit considerable variation across regions and seasons. These fluctuations may influence public attitudes and perceptions towards rodents, as communities experiencing seasonal surges in rodent populations or frequent human-rodent interactions might demonstrate heightened negative attitudes. This factor could potentially limit the generalizability of our findings to different contexts. Future research employing the SARod scale should account for such temporal and spatial variations to gain a more comprehensive understanding of how fluctuating rodent populations impact public attitudes over time and across diverse geographic areas.
4) To give, if possible, some concrete exemples of the impact of such findings in the application of strategies in different development sectors, such food management, public health, pest control regulations...
A: Thank you for this insightful suggestion. We agree that emphasizing practical applications of the findings can strengthen the relevance of the SARod scale in diverse sectors. Accordingly, we have incorporated concrete examples within the discussion section to illustrate how understanding public attitudes toward rodents can inform strategies in areas such as food management, public health, and pest control regulation. We have introduced this new paragraph:
The insights gained from the SARod scale have tangible implications for various development sectors. In the realm of food management, understanding public attitudes towards rodents can facilitate the development of more effective awareness campaigns that promote preventive measures among agricultural workers and local communities. This could lead to reduced crop losses without the need for harmful rodenticides. Within public health, the scale can identify population segments with heightened fear or misconceptions about rodents, enabling targeted health education that addresses zoonotic disease risks and promotes safe coexistence practices. Moreover, in the field of pest control regulation, recognizing the social perceptions of rodents can inform policy-making and regulatory approaches that balance ecological concerns with human health priorities. By integrating the findings of the SARod scale, these sectors can develop strategies that are culturally sensitive, reduce reliance on aggressive control methods, and foster a more sustainable approach to rodent population management.
Reviewer 3 Report
Comments and Suggestions for Authors
Dear Authors,
The manuscript certainly touches upon an interesting topic. The population has long had a negative attitude towards rodents. Some people are afraid of them, while others are very scared by the word rodent. However, the authors have not yet managed to reveal the essence of the proposed model. Firstly, it is not shown which types of rodents were most often associated by the population during surveys. This is important. Did the assessment of a mouse and a rat differ? At the beginning of the manuscript, it is necessary to describe in more detail the interest in dividing the list of rodents into useful and harmful in Chile. This is not yet the case. Secondly, the authors have missed the development of practical measures to reduce people's negative attitudes towards rodents. This is important to indicate in the Conclusion. The text of the manuscript is mixed up in different chapters and should be moved to the appropriate chapters. In many methodological aspects, the authors omit important information. It should be added. The comparative part in the discussion needs to be expanded and additional sources of literature on other countries should be cited. The authors have obtained interesting results, presented them and based on the hypotheses, they should be disclosed in the conclusions of the manuscript. After all comments have been eliminated, the manuscript can be reviewed again.

Author Response
The manuscript certainly touches upon an interesting topic. The population has long had a negative attitude towards rodents. Some people are afraid of them, while others are very scared by the word rodent. However, the authors have not yet managed to reveal the essence of the proposed model.
Firstly, it is not shown which types of rodents were most often associated by the population during surveys. This is important. Did the assessment of a mouse and a rat differ? At the beginning of the manuscript, it is necessary to describe in more detail the interest in dividing the list of rodents into useful and harmful in Chile. This is not yet the case.
A: Thank you for your thoughtful feedback. As the reviewer correctly suggests, attitudes toward rodents are likely to vary significantly depending on the rodent type. However, constructing and validating a multidimensional instrument that captures general attitudes, such as the SARod, is a critical first step. This common, validated questionnaire provides a foundation for future comparisons, allowing us to assess nuanced differences in perceptions of various rodent types (e.g., house mice versus native species) without introducing bias in initial responses. Establishing this baseline will support future research in comparing specific rodent attitudes across cultural and ecological contexts. We clarify this point in the introduction section:
Attitudes toward rodents are likely to vary significantly depending on the type of rodent, with certain species, such as house mice or native species, potentially eliciting distinct responses. However, as a necessary first step, constructing and validating a multidimensional instrument like the SARod is essential for capturing general attitudes without introducing pre-defined distinctions among rodent types. This common, validated questionnaire will serve as a foundation for future research, enabling the precise comparison of attitudes toward different rodent types across varied cultural and ecological contexts.
Secondly, the authors have missed the development of practical measures to reduce people's negative attitudes towards rodents. This is important to indicate in the Conclusion.
A: Thank you very much for your comment. We have added the following text at the end of the conclusions:
Similar to the trajectories followed by other conservation programs (e.g. Latin Ameri-can and Caribbean Network for the Conservation of Bats-RELCOM), which represent an-imals with different degrees of conservation threat and conflicts with humans, rodents must be considered as an order and as particular species. One response is to create pro-grammes or actions, adapted to the culture and legislation of each country; which con-template research, conservation and management actions; and education and dissemina-tion, in order to reduce negative human attitudes towards rodents. This requires a coex-istence approach that contemplates both positive and negative interactions, and exhaus-tive planning at different sociocultural and spatial scales (Marchinin et al. 2021). Actions associated with reducing damage (avoiding or compensating economic losses, preventing zoonoses, etc.) and monitoring population abundance have a great impact (Bencin et al. 2016)”
The text of the manuscript is mixed up in different chapters and should be moved to the appropriate chapters.
A: We do not understand exactly what this refers to. However, we hope that the changes made throughout the text to improve the clarity of the manuscript have improved this point.
In many methodological aspects, the authors omit important information. It should be added.
A: Thank you for your comments. The sections on design, participants, instruments, procedures and data analysis contain the information that is usually included in studies of this type. We hope that the changes made to the text will help to improve this point.
The comparative part in the discussion needs to be expanded and additional sources of literature on other countries should be cited.
A: We have made several changes to the Discussion, Implications and Conclusions sections. We hope that these changes have addressed the shortcomings identified. On the other hand, this debate is addressed in the introduction:
“Previous research investigating public attitudes towards rodents is relatively limited, particularly within the Latin American region, where Fitte et al.'s [22] study in La Plata, Argentina, stands out as a notable exception. This study examined social perceptions of urban rodents and their associated health risks in two contrasting neighborhoods, revealing a shared concern about rodent-borne diseases and emphasizing the need for public-centered pest management strategies. While both neighborhoods perceived rodents as a threat, El Mondongo demonstrated greater knowledge of rodent species and habitats. In Malawi, Donga et al. [23] found that farmers widely recognized rodents as agricultural pests but often lacked awareness of the associated health risks. A media campaign in the Philippines [24] successfully improved farmers' knowledge, attitudes, and practices towards environmentally friendly rodent management, leading to increased crop yields and more selective pest control. Studies in Tanzania and Ethiopia [25] revealed that while farmers acknowledged the detrimental impact of rodents on agriculture, particularly in monocultures, they often lacked the necessary information for effective management, resulting in inconsistent and ineffective control measures. Morzillo and Mertig [26] analyzed the relationship between demographic and socioeconomic variables with attitudes towards rodents in various urban contexts. Their findings revealed significant differences in attitudes, with males generally expressing more negative views than females and younger, more educated individuals tending to have more positive perceptions”
The authors have obtained interesting results, presented them and based on the hypotheses, they should be disclosed in the conclusions of the manuscript.
A: Thank you very much. We hope that the further comments in the conclusions section will respond to this point.
After all comments have been eliminated, the manuscript can be reviewed again.
Line 37-39: This is not needed here.
A: This information is removed from the abstract.
Line 46: In the world fauna, scientists are constantly revising the taxa of rodents in different geographical areas [Bryja et al. 2019, Andreychev, Kuznetsov 2020].
Thank you very much for your comment. We have added the following information:
A: Despite the order Rodentia's diversity, encompassing species as disparate as squirrels and capybaras, it is often narrowly perceived as synonymous with rats and field mice. The number of species belonging to the Rodentia order is under continuous review worldwide due to its taxonomic complexity. It is the most diverse order within mammals due to its limited dispersion and ecological specialization, contributing a large number of endemism [1,2]. In Chile, Rodentia is the most species-rich mammalian order, accounting for 67 of the country's 163 living mammal species, distributed across 7 families (Caviidae, Chinchillidae, Abrocomidae, Ctenomyidae, Echimyidae, Octodontidae, Cricetidae) and 30 genera. The Chilean rodent fauna includes iconic species such as chinchillas, coypu, and mice. There are also five introduced species—Castor canadensis, Ondatra zibethicus, Mus musculus, Rattus exulans, and Rattus norvegicus—several of which raise significant sanitary and domestic concerns due to their association with human activity [2].
Line 48: Give in brackets
A: The suggested change is incorporated
Line 50: Indicate how many species of introduced rodents there are in Chile
A: Thank you. We added the name of the families in parentheses as shown below.
“In Chile, Rodentia is the most species-rich mammalian order, accounting for 67 of the country's 163 living mammal species, distributed across 7 families (Caviidae, Chinchillidae, Abrocomidae, Ctenomyidae, Echimyidae, Octodontidae, Cricetidae) and 30 genera.”
Line 53: Specify how many species are typical for each environment. Where are the dominants.
A: Thank you. We add the following phrase on line 50 of the text:
“There are also five introduced species—Castor canadensis, Ondatra zibethicus, Mus mus-culus, Rattus exulans, and Rattus norvegicus—several of which raise significant sanitary and domestic concerns due to their association with human activity (D'Elía et al., 2020).”
Line 107-110: This is for Discussion.
A: Thank you very much for your comment. We believe it is important to mention this here as well, in order to understand the importance of the formulation of our objective.
Line 136-145: Write the aim of the research clearly.
A: The general objective and the specific objectives are formulated before the hypotheses:
“To this end, after defining the construct and developing an initial pool of items, we set the following specific objectives: (1) to conduct a descriptive analysis of the SARod items; (2) to provide evidence of validity based on the internal structure of the scale; (3) to assess the reliability of SARod through internal consistency measures; and (4) to examine the validi-ty of the SARod in relation to sociodemographic variables (gender, age, education), level of interaction with rodents, and willingness to engage in rodent extermination behaviors”
Line 153-156: This is in the Introduction
A: The design section is removed. This is the current text:
This work presents an instrumental design [30, 31]. We consider the methodological recommendations of Abad et al. [32] and Lloret-Segura et al. [33] for decision making re-garding the selection of evidence of validity, reliability and statistical analysis.
Line 159: Is this sample sufficient?
A: This is already stated later in the text, both in the data analysis and results section. Data analysis section saying:
“We also evaluated sample size adequacy, guided by communalities in relation to the number of items per factor. According to Lloret-Segura et al. [31], for sample sizes of 200 participants, communalities between 0.4 and 0.7 and at least 3 or 4 items per factor are considered acceptable.”
And in the results section saying:
“We used subsample 1 (n = 248) to explore factor structure. The KMO index of = .911 and Bartlett's Sphericity test (χ2(496) = 4738, p < .001) indicated that the correlation matrix was suitable for Exploratory Factor Analysis (EFA). With the exception of items 23 and 29, which had communalities of 0.362 and 0.378 respectively, the communalities ranged from 0.413 to 0.787, confirming that the sample size was adequate for conducting the EFA.”
Woth noting that there is no single criterion for determining the minimum sample size for factor analysis (Lloret-Segura et al., 2014; Mundfrom, 2009). Some authors propose general guidelines, suggesting minimums of 200 subjects or more (Comrey & Lee, 1992). Alternatively, others recommend ratios based on the number of participants per variable, such as 6 or 10 subjects per variable (Cattell, 1978; Everitt, 1975). However, more recent reviews highlight the importance of considering factors such as communalities and the number of items per factor, noting that higher communalities allow for smaller sample sizes (Mundfrom, 2009).
Cattell, R. B. (1978). The scientific use of factor analysis. New York: Plenum.
Comrey, A. L., & Lee, H. B. (1992). A first course in factor analysis. Hillsdale, NJ: Lawrence Erlbaum Associates, Inc
Mundfrom, D. J., Shaw, D. G., & Ke, T. L. (2005). Minimum Sample Size Recommendations for Conducting Factor Analyses. International Journal of Testing, 5(2), 159–168. doi:10.1207/s15327574ijt0502_4
Line 196-213: It is not clear to the reader what questions were included in the questionnaire. Therefore, some of the questions should be given in the text of the manuscript.
A: Thank you for this helpful suggestion. To provide clarity on the questionnaire items, sample questions are included later in the manuscript within the results section, as is customary in questionnaire validation studies. This structure allows readers to view the questions alongside the psychometric analyses and factor loadings for each item, which facilitates a better understanding of how each item contributes to the scale's validity. We hope this approach meets your expectations for transparency and detail in presenting the SARod's content.
Table 2: The rationale for using these questions should be given in the Materials and Methods.
A: Thank you very much for your comment. We added the following information
“Each of these items was designed/selected considering its representativeness and rele-vance to the theoretical construct. Adequacy was also considered, i.e. the inclusion of a sufficient number of items to represent the key aspects of the constructs, while avoiding irrelevant or redundant items”
Line 358-360: The manuscript should pay detailed attention to clarifying what species of rodents were discussed when interviewing people.
A: Thank you for your comment. The SARod was designed to assess general attitudes toward rodents as a group, without prompting respondents to focus on specific rodent species. This approach enables the scale to capture a comprehensive measure of general rodent attitudes, creating a foundation for future studies that might explore attitudes toward particular species. To clarify this, we have added a statement to the methodology section specifying that participants were asked to consider rodents in general rather than any specific species.
In designing the SARod, participants were prompted to consider rodents in a general sense, without focusing on specific species. This approach was taken to ensure that the scale captures broad attitudes toward rodents, facilitating an unbiased measure that can be applied across various contexts. By capturing general attitudes, the SARod aimed to provide a foundation for future studies that may wish to investigate attitudes toward particular rodent species or types.
Line 382-387: This part of the text is not suitable for discussion. It is more for the Introduction.
A: We deleted the following information
“Given the potential threat posed by negative attitudes toward rodent conservation and the lack of a psychometrically sound instrument to measure these attitudes in Chile, we developed the Scale of Attitudes toward Rodents (SARod)”
Line 547-550: What practical recommendations can you make based on the results of your research?
A: We have added information about this both as limitations and in the conclusions section.
References: Add:
- Bryja, J.; Meheretu, Y.; Šumbera, R.; Lavrenchenko, L.A. Annotated checklist, taxonomy and distribution of rodents in Ethiopia. Folia zoologica 2019, 68, 117–213. https://doi.org/10.25225/fozo.030.2019
- Andreychev, A.; Kuznetsov, V. Checklist of rodents and insectivores of the Mordovia, Russia. ZooKeys 2020, 1004, 129–139. https://doi.org/10.3897/zookeys.1004.57359
A: We understand the importance of including taxonomic information, but we believe it is more appropriate to include information corresponding to Chile, or failing that, Latin America.
Round 2
Reviewer 1 Report
Comments and Suggestions for Authors
I appreciate the revisions the authors have made in responding to my comments. Some very minor grammar corrections required lines 429 (property not properties) 471 (second not secondly for consistency) 504 (property not properties and the numeric 3 is redundant?).
Reviewer 3 Report
Comments and Suggestions for Authors
Dear Authors,
Yes, the manuscript has indeed improved after editing. The addition to the Introduction is certainly important. The text on Materials and Methods has also been strengthened. Clarity has been added to the Conclusion, which was not the case in the previous version of the manuscript. I have no more comments.